# Review of Contemporary Invasive Treatment Approaches and Critical Appraisal of Guidelines on Hypertrophic Obstructive Cardiomyopathy: State-of-the-Art Review

**DOI:** 10.3390/jcm11123405

**Published:** 2022-06-14

**Authors:** Steven Lebowitz, Mariusz Kowalewski, Giuseppe Maria Raffa, Danny Chu, Matteo Greco, Caterina Gandolfo, Carmelo Mignosa, Roberto Lorusso, Piotr Suwalski, Michele Pilato

**Affiliations:** 1University of Pittsburgh School of Medicine, Pittsburgh, PA 15213, USA; lebowitz.steven@medstudent.pitt.edu; 2Cardio-Thoracic Surgery Department, Heart and Vascular Centre, Maastricht University Medical Centre (MUMC), 6200 MD Maastricht, The Netherlands; lorussobs@gmail.com; 3Clinical Department of Cardiac Surgery, Central Clinical Hospital of the Ministry of Interior and Administration, Centre of Postgraduate Medical Education, 00-213 Warsaw, Poland; suwalski.piotr@gmail.com; 4Thoracic Research Centre, Collegium Medicum, Nicolaus Copernicus University, Innovative Medical Forum, 87-100 Bydgoszcz, Poland; 5Department for the Treatment and Study of Cardiothoracic Diseases and Cardiothoracic Transplantation, IRCCS-ISMETT, 90127 Palermo, Italy; graffa@ismett.edu (G.M.R.); sagreco@ismett.edu (M.G.); cgandolfo@ismett.edu (C.G.); cmignosa@ismett.edu (C.M.); mpilato@ismett.edu (M.P.); 6Department of Cardiothoracic Surgery, Division of Cardiac Surgery, University of Pittsburgh Medical Center Heart & Vascular Institute, University of Pittsburgh School of Medicine, Pittsburgh, PA 15213, USA; chud@upmc.edu

**Keywords:** hypertrophic obstructive cardiomyopathy, septal myectomy, alcohol septal ablation, left ventricle outflow tract obstruction, mitral valve surgery

## Abstract

Background: Hypertrophic obstructive cardiomyopathy (HOCM) is a heterogeneous disease with different clinical presentations, albeit producing similar dismal long-term outcomes if left untreated. Several approaches are available for the treatment of HOCM; e.g., alcohol septal ablation (ASA) and surgical myectomy (SM). The objectives of the current review were to (1) discuss the place of the standard invasive treatment modalities (ASA and SM) for HOCM; (2) summarize and compare novel techniques for the management of HOCM; (3) analyze current guidelines addressing HOCM management; and (4) offer suggestions for the treatment of complex HOCM presentations. Methods: We searched the literature and attempted to gather the most relevant and impactful available evidence on ASA, SM, and other invasive means of treatment of HOCM. The literature search yielded thousands of results, and 103 significant publications were ultimately included. Results: We critically analyzed available guidelines and provided context in the setting of patient selection for standard and novel treatment modalities. This review offers the most comprehensive analysis to-date of available invasive treatments for HOCM. These include the standard treatments, SM and ASA, as well as novel treatments such as dual-chamber pacing and radiofrequency catheter ablation. We also account for complex pathoanatomic presentations and current guidelines to offer suggestions for tailored care of patients with HOCM. Finally, we consider promising future therapies for HOCM. Conclusions: HOCM is a heterogeneous disease associated with poor outcomes if left untreated. Several strategies for treatment of HOCM are available but patient selection for the procedure is crucial.

## 1. Introduction

Hypertrophic cardiomyopathy (HCM) is the most common genetic heart disease, affecting about one in 200–500 people, but only a minority of cases (10–20%) are identified clinically [1,2]. It results from a genetic disorder with an autosomal dominant inheritance pattern caused by mutations in sarcomere proteins [3]. These mutations manifest phenotypically with myocardial hypertrophy and a small left ventricular cavity [3]. A subset of people with HCM have an obstructed left ventricular outflow tract (LVOT), which is the hallmark of hypertrophic obstructive cardiomyopathy (HOCM) [3]. One of the potential long-term effects of HOCM is heart failure (HF), and patients are at an increased risk of sudden cardiac death (SCD). Thus, intervention is critical in patients with advanced HOCM. The gold-standard treatment for HOCM is surgical septal myectomy (SM), in which a portion of the interventricular septum is removed to decrease LVOT obstruction (LVOTO) [4]. Alcohol septal ablation (ASA) is an alternative treatment for HOCM. This is a minimally invasive, intravascular procedure in which absolute alcohol is injected into the ventricular septal myocardial vasculature to induce necrosis of septal myocytes, thereby decreasing septal thickness and related LVOTO [5]. Although ASA is favorable in terms of invasiveness and recovery, SM is still considered the primary intervention for HOCM [5].

Importantly, there is a distinct advantage to invasive management over medical treatment of HOCM patients [6]. Indeed, more recently, other contributing factors in LVOTO were disclosed: the mitral valve (MV) and its subvalvular apparatus (SVA). It is noteworthy that in some patients with HOCM, the mechanism of LVOTO is entirely independent of septal hypertrophy. However, lengthened anterior mitral leaflet, bifid papillary muscles, and abnormal chordal attachment may also be causally associated with LVOTO [7]. Given this knowledge, treatment of the mitral valve and subvalvular apparatus should be considered along with SM where indicated [8].

Defining the appropriate treatment for HOCM might be challenging due to the extreme heterogeneity of the disease. SM with or without MV procedures and ASA are well recognized options for treatment [9]. The best approach, however, is still a matter of debate, and a patient-tailored approach represents a critical and necessary process.

## 2. Left Ventricular Outflow Tract Obstruction—Outcomes If Left Untreated

About two thirds of patients with HCM present with LV cavity obstruction, accounting for a diagnosis of HOCM [10]. Mechanisms of LVOTO in HOCM are many, including isolated massive septal hypertrophy; systolic anterior motion (SAM) of the MV, in which the MV leaflet contacts the septum during systole; papillary muscle (PM) anomalies; and mitral leaflet anomalies [11]. Furthermore, these mechanisms may intervene in an isolated or associated fashion. The definition of LVOTO is dynamic outflow pressure >30 mmHg; this phenomenon is typically considered hemodynamically important when pressure >50 mmHg [11].

Patients with HCM and HOCM are at an increased risk of arrhythmogenic SCD [10]. This risk of SCD in HOCM does not have a correlation with LVOT gradient severity [1] and, therefore, is not only present in symptomatic individuals but also in those who are young and asymptomatic, including young athletes [8] in whom HCM is the most common cause of SCD [1]. Implantable cardioverter defibrillators (ICDs) are an effective treatment for terminating these potentially lethal ventricular arrhythmias. Complications related to the ICD implant, however, besides procedural complications [10], include inappropriate discharge and related anxiety and mental anguish. The decision to implant an ICD requires risk stratification. Risk factors for SCD include: patient history of ventricular fibrillation, sustained ventricular tachycardia, non-sustained ventricular tachycardia, or unexplained syncope; maximum left ventricular wall thickness >3 cm; abnormal blood pressure response during exercise; and family history of SCD in the setting of HCM [12]. Recently, late gadolinium enhancement on cardiac magnetic resonance imaging, a surrogate of myocardial fibrosis, apical aneurysms, and systolic dysfunction (LVEF < 50%), have been identified as significant risk factors for SCD [11].

The 2020 AHA/ACC guidelines offer the latest and, as yet most comprehensive description of SCD risk in patients with HCM. This document provides an algorithm for risk assessment, which helps to inform providers on the necessity of ICD implantation [13]. Prior to this, the 2014 ESC guidelines offered a formula which incorporates clinical variables to generate a risk score (SCD risk score > 6% is considered high and calls for ICD implantation) [11,12].

The superiority of invasive over medical treatment has been demonstrated. Indeed, Ball et al. analyzed 649 HOCM patients with resting LVOTO and compared invasively and conservatively treated groups. This study showed significantly lower mortality at 1-, 5-, and 10 years in the invasive group compared to the conservative cohort [14]. Therefore, patients with HOCM represent a unique opportunity to decrease the risk of SCD through septal reduction therapy. Although septal reduction has yet to be indicated for the reduction of SCD risk, a growing body of evidence supports that SM and ASA do decrease risk for SCD [10].

Heart failure is a known long-term outcome of untreated and refractory HOCM [10]. Patients treated with SM have a reduction in LVOTO and excellent long-term outcomes equivalent to the general population and superior to HOCM patients treated conservatively [4,15]. ASA is also a safe and effective treatment for HOCM, with good long-term survival compared to the general population and to those treated with SM [16,17].

The LVOT gradient itself may present an alternative method of risk stratification for patients with HOCM. This concept was observed in Lu et al.’s study, which stratified patient prognosis (using a composite adverse event rate) according to LVOT gradients. They found that patients with the best prognosis had a provoked LVOT gradient between 30–89 mm Hg, while a resting LVOT gradient >30 mm Hg was associated with the worst prognosis. Patients with a provoked LVOT gradient <30 mm Hg or >90 mm Hg fell into an intermediate risk category [18].

## 3. Invasive Management: Alcohol Septal Ablation

ASA is a minimally invasive procedure with a shorter recovery time compared to SM. These factors make it an attractive option for many HOCM patients without a clear indication for SM, such as requiring a concomitant MV procedure. Several studies have demonstrated the safety and efficacy of ASA; in one retrospective analysis of 952 patients who underwent ASA, Batzner et al. reported a 95.8% 5-year survival [19]. A separate, multinational study of 1275 ASA patients determined 1-, 5-, and 10-year survival to be 95%, 89%, and 77%, respectively [17].

Despite this promising data, ASA also carries significant risks. In a smaller study of 80 ASA patients, permanent pacemaker implantation was required in almost 10% of the treated patients, 10% required a re-do ASA and 2.5% went on to require SM, despite the initial procedure providing a significant reduction in septal thickness [20]. Batzner et al. demonstrated that 10.5% of patients required placement of a permanent pacemaker at the time of ASA, 1.9% required subsequent SM, and 5.1% required pacemaker placement after ASA [19]. Another study of 91 HOCM patients who underwent ASA found that approximately one in three patients had major complications, including cardiac death. Additionally, most complications were late, indicating the long-term risks associated with ASA [21].

The above-mentioned data are relevant in evaluating ASA for the treatment of HOCM, but other factors, including provider experience, must also be considered. A retrospective analysis by Veselka et al. addressed clinical experience as a determinant of ASA outcomes [22]. This study, including 1310 ASA patients at multiple clinical centers, analyzed differences between the first 50 patients (“first-50”) and subsequent patients (“over-50”) treated at each center. Significant findings between the first-50 and over-50 groups included a reduction in major cardiovascular adverse events from 21% to 12%. Additionally, the following significant outcomes were determined for the over-50 group: less major adverse events and cardiovascular mortality; less self-reported dyspnea of New York Heart Association (NYHA) class III and IV; less likelihood of having an LVOT gradient >30 mm Hg; and less likelihood of having a repeated septal reduction therapy [22]. Another study from the Mayo Clinic demonstrated the excellent outcomes of ASA at experienced centers, showing no significant difference in survival at a 5.7-year average follow-up compared to age- and sex-matched patients who underwent SM [16].

Age is yet another important factor to consider when discussing the safety of ASA for the treatment of HOCM. An analysis of 1197 patients with HOCM who underwent ASA aimed to elucidate outcomes between patients </= 50 years old (“young”), 51–64 years old (“middle-aged”), and >/= 65 years old (“older”); it was found that young patients had a significantly lower 30-day mortality and pacemaker implantation incidence than older patients. Additionally, young patients had a significantly reduced annual mortality rate compared to middle-aged and older patients [23].

Overall, available data suggest that ASA is a safe procedure, especially in younger patients, with a reduced length of recovery compared to myectomy. Significant drawbacks to discuss with patients, however, include the relatively high risk of permanent pacemaker implantation and re-intervention either via re-do ASA or the more invasive SM. Finally, any patient seeking treatment with ASA should be referred to a highly experienced center of clinical excellence for optimal outcomes. Evidence on ASA is further gathered in Table 1.

## 4. Invasive Management: Surgical Myectomy

Surgical septal myectomy is considered the gold-standard treatment for HOCM [4]. Figure 1.

This technique is more invasive than ASA, as it most often involves median sternotomy. Once the heart is exposed, the operative technique involves removal of the hypertrophied portion of the muscular ventricular septum through an aortotomy in order to reduce the LVOTO. Figure 2.

Myectomy alone is associated with low operative mortality (<1%) and good long-term survival [24]. Septal approach without opening the aorta and approaching the IV septum through the mitral valve is a potential option (Figure 1).

The safety and efficacy of SM have been extensively validated. In a retrospective study of 93 patients with HOCM who underwent myectomy in China, NYHA functional class was significantly decreased from an average of 3.09 to 1.12. Additionally, SAM of the anterior mitral leaflet (AML) was completely resolved in 98.9% of patients. Of note, SAM was present in all patients preoperatively. The LVOT gradient was also significantly reduced. In this cohort, 37 patients underwent concomitant operations with myectomy with no operative mortality [25]. These results indicate that SM, with or without concomitant operations, is safe and efficacious when performed at a dedicated and experienced referral center. In another study, Ommen et al. reported long-term results of HCM patients. Patients were divided into three groups: those who underwent myectomy; those with LVOT obstruction who received conservative treatment; and those with non-obstructive HCM. Patients in the operative group did not have significantly different 1-, 5-, and 10-year survival compared to the non-obstructive group, nor compared to age- and sex-matched members of the general population. Additionally, operative patients had significantly better 1-, 5-, and 10-year survival compared to the conservatively treated HOCM group [4]. These findings demonstrate that SM is safe and that long-term outcomes are improved in HOCM patients who undergo myectomy compared to those treated conservatively.

A feared complication of HOCM is SCD and ventricular septal disruption. A retrospective study of 125 patients from the Mayo Clinic sought to determine the efficacy of SM in reducing fatal cardiac arrhythmia events. All patients in this study received ICD implantation and were divided into myectomy and non-myectomy groups. Among the myectomy group, only one patient experienced defibrillator discharge as opposed to 12 patients in the non-myectomy group (*p* = 0.004) over a follow-up period of about 4.5 years [26]. These results suggest that SM may reduce the likelihood of SCD in HOCM patients by reducing the rate of fatal cardiac arrhythmias after surgery.

Atrial fibrillation (AF) is a significant comorbidity in HCM, occurring in as many as 18% of patients [27]. In a recent study, Lapenna et al. sought to determine the feasibility of surgical ablation for refractory AF in HOCM patients undergoing concomitant cardiac procedures. Most patients underwent concomitant SM, while a smaller proportion of the 31-patient cohort underwent mitral valve repair or replacement. This study reported an operative mortality of 6% but a promising 7-year survival (87 +/− 6.1%). Moreover, they found that surgical ablation relieved AF at medium-term timepoints for a majority of patients, albeit also requiring medical therapy for satisfactory results [28].

It is also important to understand which patients may be poor candidates for SM. In a cohort of 503 SM patients, 19 (3.8%) were refractory to surgical intervention with a post-operative NYHA class of III-IV. While massive septal thickness (≥30 mm) and younger age (<30 y.o.) were identified as predictors of suboptimal response to surgery [29] due to persistent or recurrent heart failure postoperatively, most young patients and those with massive hypertrophy experienced significant symptom relief and sustained clinical improvement from SM [29].

A wealth of data vouches for the efficacy and safety of SM. Patients who undergo this procedure have excellent short- and long-term outcomes. These include low operative mortality, decreased NYHA class, excellent long-term survival, and decreased event rates for arrhythmia. Thus, SM should be considered as the primary intervention for HOCM when performed by experienced operators. Table 2 summarizes the evidence on SM in the treatment of HOCM.

## 5. Surgical Myectomy vs. Alcohol Septal Ablation: Meta-Analyses

To date, no randomized controlled trials have compared SM and ASA. However, several meta-analyses have made efforts to compare these procedures in a head-to-head fashion. The studies report, for the most part, roughly similar findings. They agree that SM and ASA provide similar: improvement in NYHA class [30,31,32]; in-hospital [31], short-term [32], and long-term [9,32,33,34] mortality; and rates of SCD [9,33,34]. Many of the meta-analyses also conclude that ASA is associated with significantly greater rates of permanent pacemaker implantation [9,31,32,33,34] and re-intervention [9,33,34]. Interestingly, the latest meta-analysis (Osman et al., 2019) reported significantly increased rates of peri-procedural mortality and stroke in patients who underwent SM [9]. The seven meta-analyses discussed in this review are described in Table 3.

The analyses conducted by Zeng et al. [30] and Alam et al. [31] included only three and five studies, respectively. Taking this into account, along with their relatively older publication dates, these meta-analyses may be considered less robust than their counterparts. Conversely, the meta-analysis by Osman et al. [9] is not only the latest, but also the largest (40 studies included with a total of 4240 patients) analysis to-date. This study asserts that ASA is associated with more ICD implantation and greater rates of re-intervention, whereas SM is associated with greater rates of peri-procedural mortality and stroke [9]. The former claim is supported by older literature [9,31,34], but the latter is novel and warrants further investigation.

Baseline patient characteristics must also be considered when comparing ASA and SM. Three of the meta-analyses found that patients who underwent SM were significantly younger than those who were treated with ASA [31,33,35]. After adjusting for baseline patient characteristics (LVOT gradient, age, sex, NYHA class, septal wall thickness, and risk factors for SCD), Leonardi et al. reported a lower odds ratio for the effect of ASA on all-cause mortality and SCD compared with SM [35]. These findings underscore the importance of considering factors that may affect which patients are selected for which procedure. For example, patients who undergo SM may have greater disease morbidity than those who opt for ASA, contributing to increased mortality in SM patients. Table 3 summarizes the data on previous meta-analyses comparing ASA vs SM for HOCM.

## 6. Alternative Invasive Treatment Options for HOCM

Dual-chamber (DDD) pacing has been considered as an alternative approach for the treatment of HOCM. Several studies have found that DDD pacing significantly reduces the LVOT gradient at immediate, short-term, and long-term time points [36,37,38,39,40,41,42]. Almost all of these studies, however, found that DDD pacing does not reduce the LVOT gradient below 30 mm Hg [37,38,39,40,41]. Additionally, Yue-Cheng et al. found that DDD pacing significantly reduces SAM at 1–4 years post-implantation [37].

Other important measures to consider are ventricular septal reduction; NYHA class and quality of life; and mitral regurgitation (MR). Several studies have found that DDD pacing offers no significant reduction in septal thickness [36,37,43]. Extensive evidence does suggest, however, that DDD pacing offers a significant reduction in NYHA class [36,38,39,41,42,44] and improvement in other quality of life scores [39,40,41,44]. Finally, Pavin et al. found that DDD pacing can significantly improve the degree of MR in select patients. The patients in this study whose MR was refractory to pacing also experienced minimal reduction in the LVOT gradient or had non-AML elongation abnormalities (including MV prolapse or annulus calcification) [45].

Despite promising evidence for the use of DDD pacing in HOCM, it remains inferior to SM and ASA. In a comparative study of 39 patients (20 SM; 19 DDD pacing), Ommen et al. found that SM is superior to DDD pacing in reducing LVOT gradient; providing symptomatic improvement; increasing exercise duration; and increasing maximal oxygen consumption [46]. Two studies comparing DDD pacing with ASA demonstrated that these therapies provide a similar reduction in LVOT gradient [43,44]. It was found, however, that ASA is superior in reducing NYHA class [42] and reducing septal thickness [43].

The data presented above suggest that DDD pacing is capable of reducing the LVOT gradient and SAM, as well as improving MR and NYHA class. Importantly, DDD pacing provides no significant reduction in septal thickness and is inferior to SM in improving hemodynamic and functional measures and to ASA in improving NYHA class and reducing septal thickness. Pending stronger evidence, there may be a role for DDD pacing in patients with HOCM for whom SM and ASA are contraindicated and who have mild LVOT gradients and limited septal hypertrophy.

Another novel approach to the treatment of HOCM is radiofrequency catheter ablation (RFCA). This is a percutaneous procedure in which radio waves (as opposed to alcohol, as in ASA) are used to create an area of necrosis in the ventricular septum. This may provide a more targeted approach than ASA, since limitations of the latter include anatomic variability of septal perforator arteries in 5–15% of patients [47], complete heart block, and induction of arrhythmia [48].

Studies have shown that RFCA is capable of significantly reducing the LVOT gradient and NYHA class at acute [49,50], medium-term [47,48,50], and long-term [48,50,51] time points. Liu et al. found that RFCA resulted in an initial increase in septal thickness followed by a significant decrease at 1, 3, and 6 months [50]. They also reported a significant reduction in MR [50]. Another study demonstrated a significant reduction in septal thickness at 6 months [47]. Important complications of RFCA have been reported in several studies, including transient pulmonary edema in one of seven patients [48] and complete AV block requiring permanent pacemaker implantation in four of nineteen patients [49].

Currently, there are not enough data to support the use of RFCA in the treatment of HOCM. The few small studies published on this subject, however, provide preliminary support for the safety and efficacy of RFCA. If future, larger studies support these early results, there may be a role for RFCA in the treatment of HOCM patients who are poor surgical candidates and have poor vascular anatomy that is incompatible with ASA. There are no comparative studies of RFCA and ASA, and inferences regarding safety in terms of arrhythmias or postoperative PPM implantation are vague.

## 7. The Mitral Valve

The mitral valve is classically implicated in the pathophysiology of HOCM via SAM of the AML. SAM can also occur, however, due to involvement of an elongated posterior mitral leaflet [52]. The mitral valve and subvalvular apparatus may contribute to HOCM through SAM-independent mechanisms. Figure 3.

These anatomic abnormalities include abnormally long MV leaflets (anterior or posterior), bifid PM, and papillary muscle attachment directly to the mitral leaflet base. These anomalies can cause LVOTO even in the absence of septal hypertrophy [6]. Figure 4.

Surgical approaches to the treatment of the above-stated abnormalities include myectomy with concomitant MV replacement (MVR) or repair (MVr) with or without papillary muscle repositioning (Figure 5).

Many studies have validated the safety and efficacy of concomitant surgical reduction of MR with SM [52,53,54,55,56,57,58,59,60,61]. Given the inherent long-term effects of MVR (anticoagulation with a prosthetic valve, short life of a tissue valve), MVr may be favorable when possible. A prospective randomized study found that patients who underwent MVr with SM, as opposed to MVR with SM, had significantly greater overall survival at 2 years and less thromboembolic events; all other outcomes were similar [62]. Additionally, a quantitative meta-analysis of 23 studies found that MVr is superior to MVR in terms of reoperation and thromboembolic events. This meta-analysis concluded that MVr should be the first line treatment over MVR in HOCM patients with MR undergoing concomitant SM [63]. These studies fail, however, to address patients on an individual basis. In a retrospective study of 115 patients who underwent SM with either MVR (N = 48) or MVr (N = 67), Kaple et al. underscored the importance of anatomic variability in operative technique. They found that MVr is a durable method but note its limited use in patients with appropriate anatomical anomalies, such as long leaflets and degenerative MV pathology; such patients may comprise as much as half of the population of interest [64].

Some HOCM patients exhibit minimal septal hypertrophy (<18 mm); the primary mechanism of LVOTO in these patients is related to MV pathology. In these patients, myectomy is sometimes forgone in favor of MVR due to fear of iatrogenic ventricular septal defect (VSD) [65]. Two studies have demonstrated, however, that SM with or without concomitant MV intervention is safe in this population [65,66].

It is important to consider MV anomalies when selecting between SM and ASA. Studies have found that SM with concomitant MV intervention is superior to ASA in reducing SAM and MR in appropriately selected patients [67,68]. Additionally, SM with concomitant MV intervention can be performed safely and efficaciously through a minimally invasive trans-mitral approach. Studies have demonstrated reduced SAM and LVOT gradients via these techniques [69,70]. Figure 5.

Myectomy with concomitant MVR or MVr is safe and efficacious and should be performed in patients with MR and/or MV anatomical anomalies contributing to HOCM. The choice of MVR versus MVr should be made on a patient-by-patient basis and consider the individual’s anatomy. Anatomy favoring MVr includes lengthened MV leaflet(s), degenerative valve disease, and subvalvular morphologies including anomalous PM insertion and chordal attachment at the valve base [64]. This, again, underscores the importance of careful evaluation of preoperative imaging. MVr should be considered over MVR when possible due to the inherent implications associated with exogenous valves. In addition, mild septal hypertrophy does not necessarily preclude SM with or without concomitant MV intervention. These patients should be considered for the correct procedure on an individual basis without strict exclusion of SM due to fear of creating a VSD. In addition, in patients with intrinsic MV pathology, SM, with or without concomitant MV procedure, is favorable over ASA. It has also been demonstrated that minimally invasive options are safe when combining SM with MV intervention. 

## 8. The Subvalvular Apparatus

Papillary muscle (PM) morphology can contribute as a mechanism of LVOTO in HOCM [71] and particularly in patients with a minor degree of septal hypertrophy (<15 mm) [72]. This may be readily identified via echocardiography and must be considered in preoperative evaluation [73]. Numerous studies have validated the safety and efficacy of SM with concomitant PM realignment, excision of anomalous PM, and cutting of abnormally thickened chordae tendinae [74,75,76,77,78,79]. Figure 6.

Ferrazi et al. have demonstrated that surgical treatment addressing septal hypertrophy and valvular defects is effective in decreasing LVOTO, thus relieving HF symptoms while also avoiding later MV replacement [74].

A prospective randomized study found that SM with concomitant subvalvular intervention was superior in terms of abolishing LVOTO and improving MR as compared to SM alone in patients with subvalvular pathology [80].

Performing SM with a concomitant subvalvular procedure is a safe and efficacious operative strategy. Preoperative echocardiograms must be carefully evaluated to determine if subvalvular morphology requires intervention.

## 9. Mid-Ventricular and Apical Hypertrophy

Left ventricular hypertrophy (LVH) in HOCM is typically localized to the subaortic portion of the septum. Rarely, hypertrophy may be present in the mid-ventricular or apical septum [81]. Although rare, mid-ventricular obstruction is a very serious phenotype of HOCM as it has been identified as a predictor of adverse outcomes, including sudden death and potentially lethal arrhythmias [82].

Patients with mid-ventricular or apical hypertrophy often require unique operative approaches. The transapical approach is a relatively new operative technique that can be applied in cases of mid-ventricular and apical hypertrophy. A study of 113 patients with apical hypertrophy who underwent transapical myectomy reported acceptable mortality and survival and a 76% clinical improvement [83]. Several studies have found that patients with mid-ventricular obstruction can be safely and efficaciously treated with transapical myectomy, transaortic myectomy, or a combination of the two procedures. Findings included adequate survival and short- and long-term outcomes; improvement in NYHA class; and improvement in the LVOT gradients [84,85,86].

Patients with mid-ventricular and apical hypertrophy should be given special consideration. Use of transapical myectomy has provided a successful alternative to heart transplant in patients with apical hypertrophy. The transaortic approach can be extended for use in patients with mid-ventricular obstruction where applicable or can be combined with the transapical approach when obstruction extends distally.

## 10. ECMO and Other MCS

There is scant literature on the use of ECMO in the setting of HOCM. Case reports [87,88,89] demonstrate that in patients in hemodynamic crisis pre- and/or post-operatively, ECMO may be considered and used as bridging therapy until support can be withdrawn (Table 4).

Several reports describe the use of mechanical circulatory support (MCS) systems in the setting of HOCM [90,91], mostly as a bridge-to-transplantation. These treatment strategies include percutaneous interventions with interatrial shunts, left atrial assist devices (LAADs), and ventricular assist devices (VADs) in various configurations [90], but the data is limited to single-heart transplantation excellence centres [92].

## 11. Guidelines

Several guidelines exist for the treatment of HOCM and fall into two categories: guidelines for cardiac pacing, and guidelines specifically addressing HCM. The 2008 ACC/AHA/ARS guidelines and 2013 ESC guidelines are the most recent to address pacing [93,94]. In regard to HOCM, both provide very similar recommendations. They agree that DDD pacing is not typically a stand-alone interventional treatment for HOCM, but is indicated in symptomatic patients who cannot be considered for or do not wish to undergo SM or ASA [93,94].

The most recent guidelines specifically addressing the treatment of HCM are the 2011 ACCF/AHA guidelines; the 2014 ESC guidelines; and the 2020 AHA/ACC guidelines [11]^,^ [13,95]. Each represents fairly comprehensive recommendations for the invasive treatment of HOCM. They each include recommendations for SM, ASA, and DDD pacing and advise considering anatomical variants such as MV and subvalvular apparatus involvement [11,95]. The 2014 and 2020 guidelines also include recommendations for the treatment of patients with mid-cavity or apical hypertrophy [11,13].

The 2020 AHA/ACC guidelines [13] represent the most comprehensive recommendations for the invasive management of HOCM. These guidelines are more current and slightly more comprehensive than the 2011 and 2014 guidelines [11,95]. When considering DDD pacing in HOCM, the 2021 ESC pacing guidelines [96] are the most current and specific to HOCM. In addition to pacing for the management of LVOTO, they provide guidance on pacemaker implantation following ASA and SM, as well as on cardiac resynchronization therapy in end-stage HCM. Figure 7 delineates a possible algorithm on poor surgical candidate management. Given the amount of literature that has only recently been published on the subject of HOCM, current guidelines may be insufficient to account for every scenario, warranting an update.

## 12. Genetics

HCM is a genetic disease associated with autosomal dominant inheritance of mutant sarcomere proteins [3]. As a genetic disease, there is much interest in the use of genetic testing to aid in risk stratification and disease management and in the use of gene therapy as a treatment modality.

Genetic testing is often performed on patients with HCM/HOCM and, in some cases, their asymptomatic relatives. Subsequent genetic counseling aids patients in family planning and can also aid providers in risk stratification. For example, certain mutations and complex genotypes are associated with greater risk and more severe phenotypes [97]. A retrospective analysis of 626 patients with HCM aimed to determine differences in genotype positive versus genotype negative patients. Positive status was associated with more non-sustained ventricular tachycardia, history of syncope, and greater LVH, and was a risk factor for all-cause mortality, cardiovascular mortality, heart failure mortality, and SCD and aborted SCD. Negative status was associated with a higher NYHA class and greater LVOT gradient [98]. 

Studies have yielded novel and important information on the genetics of HCM, such as the impact of environmental factors on phenotype [99]. Gene therapy is an attractive, potentially curative option for the treatment of HCM. Promising data have come from a study investigating the mutant MYCB3 gene associated with infant HF and death. This study has shown promise in murine models and in human pluripotent stem cell-derived cardiomyocytes using an AAV-9 vector which transfers functional MYCB3 to restore function [100].

The global COVID-19 pandemic has raised concerns regarding the effect of infection on the heart. Bos et al. demonstrated that cardiac tissue from HCM patients exhibits a five-fold greater expression of ACE-2 than control cardiac tissue. Since the SARS-CoV-2 virus uses the ACE-2 receptor for entry into host cells, up-regulation of this protein in HCM (and possibly in other heart diseases) may provide an explanation for cardiac patients demonstrating an increased risk of infection and poor outcomes in COVID-19 illness [101].

## 13. Medical Therapy

While critical appraisal of pharmacological treatments is far beyond the scope of this review [102], HOCM-associated hypercontractility is targeted by old and novel drugs alone or in combination with invasive approaches. Among them, beta-blockers and calcium channel blockers have for years been a mainstay of therapy to reduce dynamic LVOTO, improve symptoms, and prevent atrial and ventricular arrhythmias [11,103,104,105,106]. Disopyramide (an antiarrhythmic class IA agent) is often used on top of beta-blockers to improve symptoms and reduce intraventricular gradients in patients with LVOTO due to its negative inotropic effect [107].

Several emerging treatments are currently being tested in clinical trials; perhexiline, a potent carnitine palmitoyl transferase-1 (CPT-1) inhibitor, improves myocardial energetics in HCM [108], and has the potential to reduce LVH in HCM [NCT04426578]; ranolazine, a late sodium current inhibitor was tested in RESTYLE-HCM and associated with a reduction in 24-h burden of premature ventricular complexes in HOCM [109].

A recently published randomized, double-blind, placebo-controlled, phase 3 clinical trial assessed the use of mavacamten, a cardiac myosin inhibitor, for the treatment of HOCM. Olivotto et al. reported a significant improvement in NYHA class, exercise tolerance, LVOTO, and symptom scores in patients assigned to receive mavacamten over those in the placebo group [110]. Additionally, the VALOR-HCM trial showed that mavacamten improved symptoms and significantly reduced eligibility for SM among symptomatic patients with obstructive HCM who were candidates for SM on maximally tolerated medical therapy [111,112]. These landmark clinical trials pave the way for a novel medical treatment in patients with HOCM.

## 14. Minimally Invasive Surgery

Minimally invasive SM with or without concomitant MV intervention is an attractive option for patients as it provides the outcomes of SM with less surgical injury. Case reports have demonstrated success using minimally invasive approaches such as SM with aortic valve replacement through right mini-thoracotomy [113] and robotic SM with MVr in a patient with idiopathic hypertrophic subaortic stenosis [114]. Although the latter patient did not have per definition HOCM, this case demonstrated a successful operative technique that may be employed for SM.

A retrospective analysis compared 24 patients who underwent SM via right mini-thoracotomy with 26 patients who underwent traditional SM via sternotomy. The groups had similar aortic cross clamp times, post-operative pacemaker implantation rates, LVOT gradient reduction, and residual SAM [115]. Another retrospective analysis of 34 full sternotomy and 86 mini-sternotomy SM cases reported excellent results. Both groups had significant reductions in NYHA class, similar resting LVOT gradients at follow-up, similar median times on bypass, and similar major complication rates. The mini-sternotomy group had a slightly longer (39 min versus 35 min; *p* = 0.017) time while on cross clamp [116].

A recent, larger study reported outcomes for 51 HOCM patients who underwent right mini-thoracotomy for minimally invasive septal reduction. Jiang et al. report a significant reduction in LVOT gradient and septal thickness, as well as the abolishment of SAM and mitral regurgitation (or insignificant MR) for all patients [117]. These results further underscore the feasibility and safety of minimally-invasive surgery for septal reduction and MR for patients with HOCM.

## 15. MitraClip^TM^

Percutaneous MV “edge-to-edge” repair using MitraClip^TM^ is a relatively new, minimally-invasive option for patients who are poor surgical candidates [118]. Some small studies have reported outcomes of MitraCip^TM^ use in HOCM patients. One study consisting of five HOCM patients who received the device reported reductions in SAM, LVOT gradient, MR, and NYHA class [119].

Despite little data existing on the use of percutaneous MV plication in HOCM patients, the results of small studies are impressive and indicate a role for the use of MitraClip^TM^ in the treatment of patients with HOCM who are, again, poor SM and ASA candidates.

## 16. Conclusions

Hypertrophic obstructive cardiomyopathy is a heterogeneous disease with different clinical presentations, albeit producing similar dismal long-term outcomes if left untreated. Several approaches are available for treatment of HOCM; alcohol septal ablation and surgical myectomy are safe and effective, but patient selection for the procedure is crucial. In the case of elevated operative risk, novel treatments are available and are currently being tested in the setting of HOCM.

## Figures and Tables

**Figure 1 jcm-11-03405-f001:**
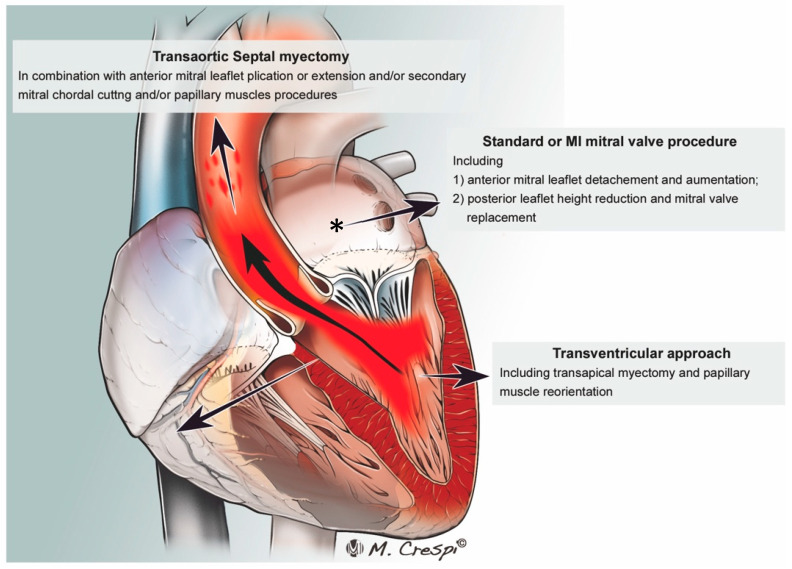
Summary of the proposed techniques for LVOT obstruction surgery. LA, left atrium. *—Conventional and minimally invasive mitral valve surgery approaches.

**Figure 2 jcm-11-03405-f002:**
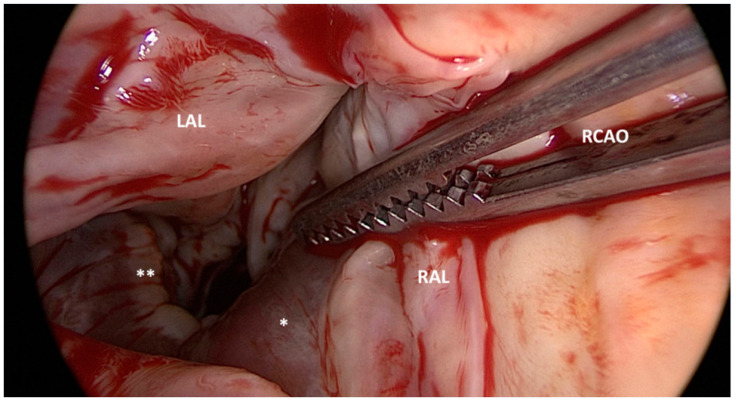
Surgical view of left ventricle outflow tract (LVOT) obstruction and the bulging of the interventricular septum (*) after left and right aortic valve leaflet retraction (LAL and RAL). RCAO: right coronary artery ostium. Jet lesions (**) produced by turbulence in the LVOT.

**Figure 3 jcm-11-03405-f003:**
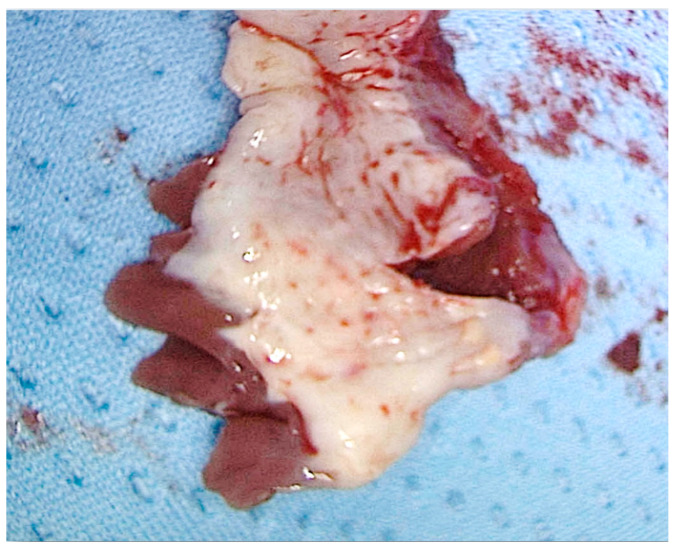
Specimen of interventricular septum showing endocardial fibrosis secondary to trauma caused by systolic anterior motion of the anterior leaflet of the mitral valve.

**Figure 4 jcm-11-03405-f004:**
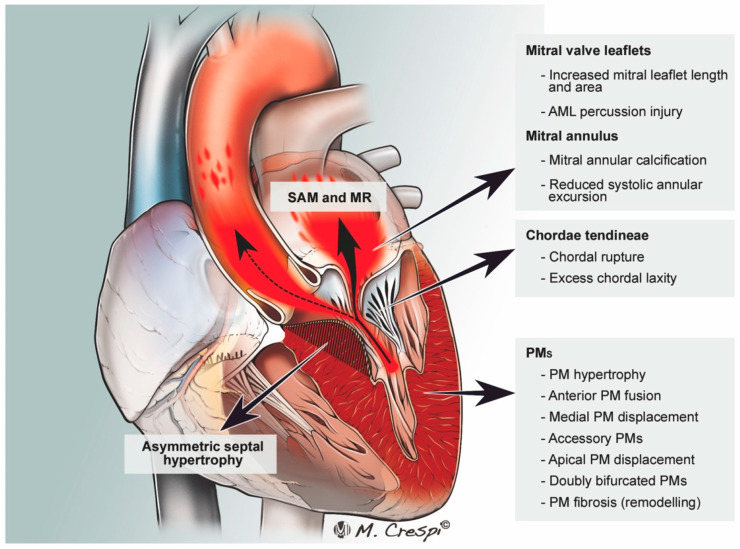
Pathophysiology of left ventricle outflow tract (LVOT) obstruction (dotted arrow) in hypertrophic obstructive cardiomyopathy. SAM: systolic anterior motion; MR: mitral regurgitation, MV, mitral valve; AML, anterior mitral leaflet; PM, papillary muscle; APM, anterior papillary muscle; MPM, medial papillary muscle. (from Silbiger J.J. et al. J Am Soc Echocardiogr 2016).

**Figure 5 jcm-11-03405-f005:**
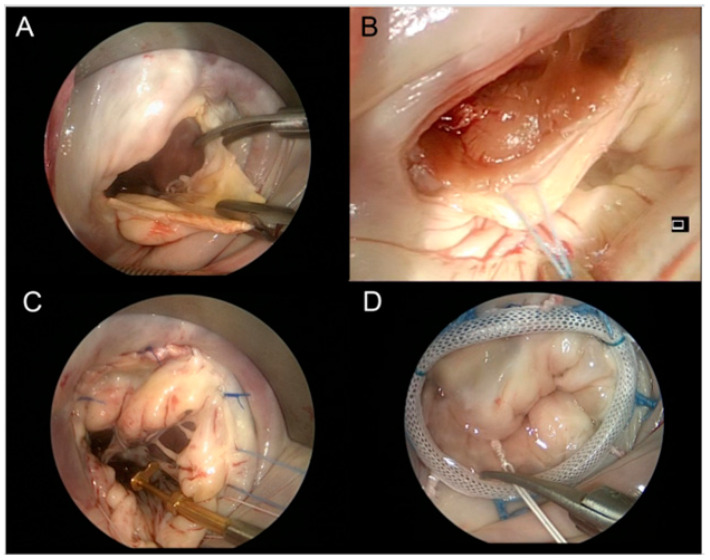
Minimally invasive HOCM surgery. Transmitral SM. Incision at the base of mitral leaflet (**A**); pull back suture placement to facilitate SM (**B**); mitral valve repair with loop-technique (**C**); completed mitral valve annuloplasty (**D**) concomitant to LVOTO repair.

**Figure 6 jcm-11-03405-f006:**
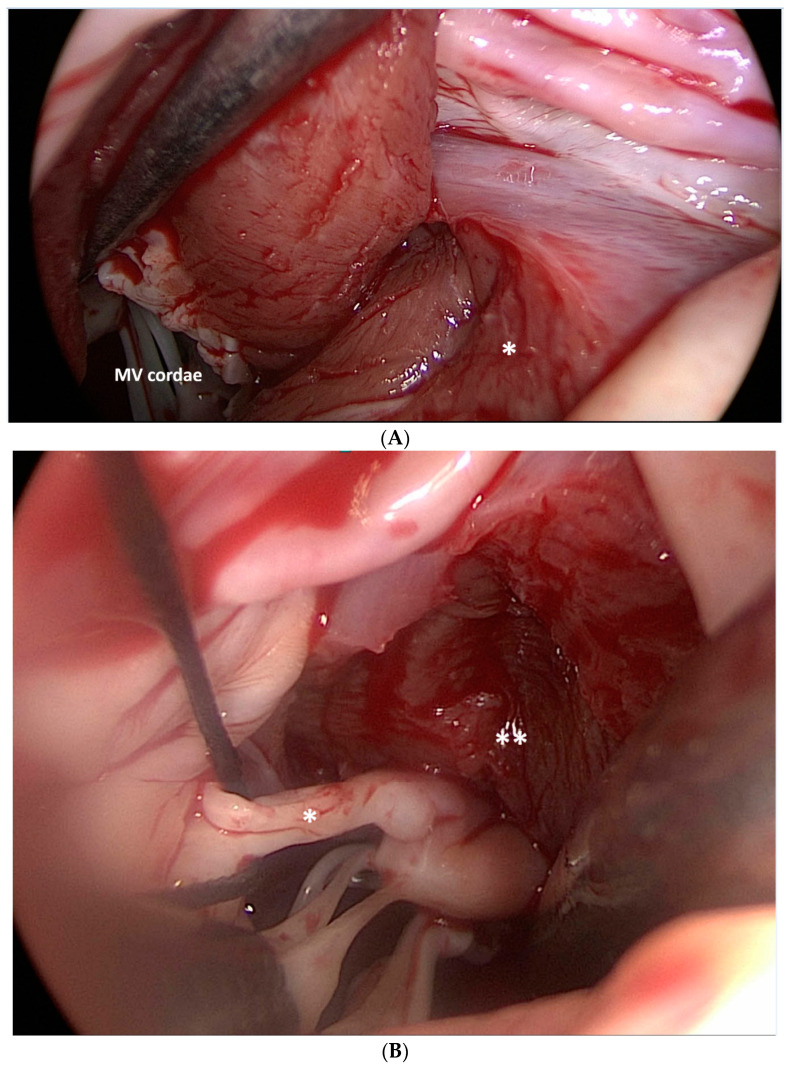
Surgical myectomy (*****) was performed starting at the nadir of the right coronary sinus, and extended apically to achieve the exposure of the papillary muscles. MV: mitral valve (**A**). Surgical view of the diseased mitral valve secondary cordae (*) and papillary muscles (**) after myectomy. It is noteworthy that the bases of the papillary muscles are now visible (**B**).

**Figure 7 jcm-11-03405-f007:**
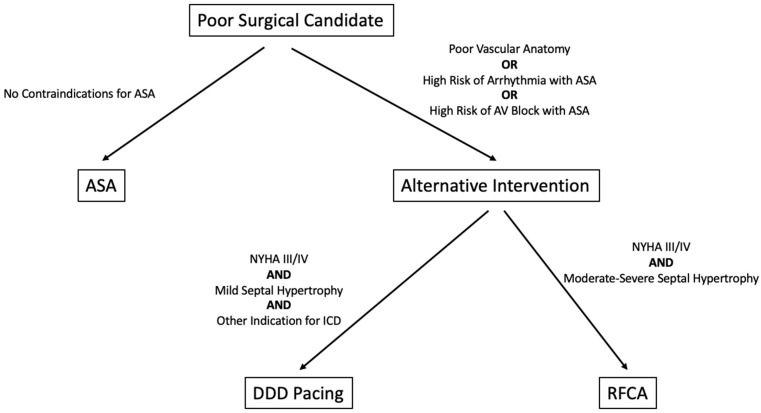
Proposed algorithm for alternative treatments when surgery is contraindicated. At this time, DDD pacing and RFCA are not indicated for the treatment of HOCM by any guidelines.

**Table 1 jcm-11-03405-t001:** Summary of the salient characteristics and results of studies on alcohol septal ablation for the treatment of hypertrophic obstructive cardiomyopathy discussed in this review.

Authors	Institution	N (Total)	Symptomatic Status Pre-ASA	N (Patients with Pre-ASA MR)	Average Pre-ASA LVOT Gradient (mm Hg)	Average Post-ASA LVOT Gradient (mm Hg)	Major Outcomes
Batzner et al. [19]	KWM Standort Juliusspital, Germany	952	698 patients NYHA Class III/IV	N/A	63.9 +/− 38.2	33.6 +/− 29.8	- Significant reduction in LVOT gradient - Estimated 5-year survival of 98.5% - 10.5% permanent pacemaker at time of ASA - 1.9% subsequent SM - 5.1% permanent pacemaker later on
Veselka et al. [17]	10 tertiary invasive European centers	1275	Average NYHA Class 2.9 +/− 0.5	N/A	67 +/− 36	16 +/− 21	- 30-day mortality of 1% - 1-, 5-, 10-year survival of 98%, 89%, 77% (respectively) - Independent predictors of all-cause mortality: pre-ASA age, septal thickness, and NYHA class; and LVOT gradient at last f/u - Significant improvement in NYHA class and LVOTG
Aguiar et al. [20]	Santa Maria Hospital, Lisbon, Portugal	80	74 patients NYHA class III/IV	26 (moderate MR)	96.3 +/− 34.6	27.1 +/− 27.4 (successful); 58.2 +/− 16.6 (unsuccessful)	- 6.3% minor complications; 2.5% major complications; 8.8% permanent pacemaker - 85.7% of patients achieved >50% reduction in LVOT gradient (successful) - 77% of patients with NYHA III/IV experienced reduction to NYHA I/II
ten Cate et al. [21]	Erasmus University Medical Center Rotterdam, Netherlands	91	91 patients NYHA class III/IV	MR grade 1.5 +/− 0.9	92 +/− 25	8 +/− 17	- 1-, 5-, 8-year survival of 96%, 86%, 67% (respectively) for ASA - 1-, 5-, 8-year survival of 100%, 96% 96% (respectively) for SM - ASA carried ~5-fold increased risk of composite cardiac death and aborted SCD compared to SM
Veselka et al. [22]	Euro-ASA Registry, 11 European Centers	1310	1098 patients NYHA class III/IV	N/A	73.9 +/− 41.8 (“first-50” group); 66.8 +/− 34.5 (“over-50” group)	20.8 +/− 27.5 (“first-50” group); 14.0 +/− 17.2 (“over-50” group)	- 30-day CV death rate of 2.1% for first-50, 0.4% for over-50 (*p* = 0.01) - 30-day pacemaker implantation rate of 15% for first-50, 9% for over-50 (*p* < 0.01) - Significantly greater rates of major adverse events and CV death in long-term f/u for first-50 group - Significantly greater rates of NYHA class III/IV, LVOT gradient > 30 mm Hg, and re-do septal reduction for first-50 group
Sorajja et al. [16]	Mayo Clinic, USA	177	177 patients NYHA class III/IV	N/A	70 +/− 40	85 +/− 16% reduction in LVOT gradient	- No significant difference in survival in ASA compared to general population and SM
Liebregts et al. [23]	7 tertiary invasive European centers	1197	NYHA class III/V by age group: 298 patients </= 50 years; 352 patients 51–64 years; 363 patients >/= 65 years	N/A	Age </= 50 years: 110 +/− 39; Age 51–64 years: 111 +/− 44; Age >/= 65 years: 121 +/− 47	Age </= 50 years: 26 +/− 31; Age 51–64 years: 27 +/− 35; Age >/= 65 years: 26 +/− 33	- Significantly lower mortality and ICD implantation in young vs. older patients - Similar adverse arrhythmic event rates among groups - Annual mortality rates of 1%, 2%, and 5% for young, middle-aged, and older patients, respectively (*p* < 0.01) - For young patients, age, residual LVOT gradient, and female sex were independent predictors of mortality

ASA, alcohol septal ablation; LVOT, left ventricle outflow tract; SM, surgical myectomy; MR, mitral regurgitation; SCD, sudden cardiac death; CV, cardiovascular; ICD, implantable cardioverter-defibrillator; N/A, not available.

**Table 2 jcm-11-03405-t002:** Summary of the salient characteristics and results of studies on surgical myectomy for the treatment of hypertrophic obstructive cardiomyopathy discussed in this review.

Authors	Institution	N (Total)	Symptomatic Status Pre-SM	N (Patients with Pre-SM MR)	Average Pre-SM LVOT Gradient (mm Hg)	Average Post-SM LVOT Gradient (mm Hg)	Concomitant Procedure(s)	Major Outcomes
Wang et al. [25]	National Center for Cardiovascular Diseases, Beijing, China	93	80 patients NYHA class III/IV	32 (mild); 30 (moderate); 10 (moderately severe); 1 (severe)	91.76 +/− 25.08	14.78 +/− 14.01	10 MVR 9 MVr 6 AVR 2 TV plasty 18 CABG 3 modified Maze 2 cardiac tumor resection 1 RVOT reconstruction 12 multiple	- Significant reduction in NYHA Class - Complete resolution of SAM in 98.9% - Significant reduction in LVOT gradient - 0% operative mortality
Ommen et al. [4]	Mayo Clinic, USA	1337 (289 SM; 228 non-operative; 820 non-obstructive HCM)	348 patients NYHA III/IV (256 SM; 34 non-operative; 58 non-obstructive HCM)	71 (21 SM; 24 non-operative; 26 non-obstructive HCM)	29.2 +/− 39 (67.3 +/− 41 SM; 68.0 +/− 31 non-operative; 5.1 +/− 7 non-obstructive HCM)	3 +/− 8 (SM group)	64 patients	- 0.8% operative mortality - 1-, 5-, and 10-year post-SM survival similar to non-obstructive HCM and general population - Survival benefit for SM over non-operative
McLeod et al. [26]	Mayo Clinic, USA	125	48 patients NYHA III/IV (27 SM; 21 non-SM)	N/A	59 +/− 35 (SM group)	1 +/− 3 (SM group)	N/A	- 12 non-SM patients vs. 1 SM patient sustained ICD discharge to prevent SCD during f/u
Lapenna et al. [28]	Vita-Salute San Raffaele University, Milan, Italy	31	17 patients NYHA III/IV	12	56 +/− 31.8	N/A	Surgical ablation with SM (77%) and/or MVR/MVr (39%)	- 6% hospital mortality - 87 +/− 6.1% 7-year survival - 1- and 6-year arrhythmia control rates of 96 +/− 3.5% and 80 +/− 8.1% (respectively)
Wells et al. [29]	Tufts Medical Center, USA	503	503 patients NYHA III/IV	34	61 +/− 38	N/A	N/A	- 96% improvement to NYHA I/II - Non-responders to SM were younger with greater extent of septal hypertrophy

SM, surgical myectomy; LVOT, left ventricle outflow tract; MVR, mitral valve replacement; MVr, mitral valve repair; AVR, aortic valve replacement; CABG, coronary artery bypass grafting; RVOT, right ventricle outflow tract; SAM, systolic anterior motion; HCM, hypertrophic cardiomyopathy; SCD, sudden cardiac death; CV, cardiovascular; ICD, implantable cardioverter-defibrillator; N/A, not available.

**Table 3 jcm-11-03405-t003:** Meta-Analyses comparing septal reduction therapies (ASA and SM).

Authors	Year	N (Total)	N (ASA Patients)	N (SM Patients)	N (Studies Included)	Outcome
Zeng et al. [30]	2006	177	86	91	3	Both ASA and SM provide LVOT gradient- and clinical improvement, more PPM following ASA.
Alam et al. [31]	2009	351	183	168	5	Both procedures safe, slightly higher LVOT gradients following ASA.
Agarwal et al. [32]	2010	708	410	298	12	Higher LVOT gradients reduction following SM, similar safety and resolution of clinical symptoms.
Leonardi et al. [35]	2010	4094	2207	1887	27	Low rates of mortality and SCD after both ASA and SM; adjusted odds ratios for SCD lower in ASA;
Liebregts et al. [34]	2015	4804	2013	2791	24	Higher rates of PPM and reinterventions following ASA; no differences in long-term
Singh et al. [33]	2016	1824	805	1019	10	Higher rates of PPM and reinterventions following ASA; no differences in short and long-term
Osman et al. [9]	2019	8453	4213	4240	40	ASA associated with lower periprocedural mortality and stroke but higher rates of PPM and reintervention, no differences in long-term

ASA, alcohol septal ablation; SM, surgical myectomy; LVOT, left ventricle outflow tract; PPM, permanent pacemaker; SCD, sudden cardiac death.

**Table 4 jcm-11-03405-t004:** Salient characteristics and results of ECMO case reports discussed in this review.

Authors	Institution	Indication for ECMO	Outcomes
Husaini et al. [87]	Washington University School of Medicine, USA	Cardiogenic shock secondary to Takotsubo cardiomyopathy	V–A ECMO until patient stable enough for SM
Basic et al. [88]	Kerckhoff Heart and Thorax Center, Germany	Cardiogenic shock	ECMO until patient stable enough for SM with MVR
Williams et al. [89]	Prince Charles Hospital, Australia	Chronic thromboembolic pulmonary hypertension	ECMO pre- and post-operatively (pulmonary endarterectomy, SM, and MVR)

V–A ECMO, veno-arterial extracorporeal membrane oxygenation; SM, surgical myectomy; MVR, mitral valve replacement.

## Data Availability

Data underlying this article will be shared on reasonable request to the corresponding author.

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
