# Peer review of "Review of Contemporary Invasive Treatment Approaches and Critical Appraisal of Guidelines on Hypertrophic Obstructive Cardiomyopathy: State-of-the-Art Review"

_jcm, 2022, doi:10.3390/jcm11123405_

Round 1
Reviewer 1 Report
In this Review entitled « Review of Contemporary Treatment Approaches and Critical Aprisal of Guidelines on Hypertrophic Obstructive Cardiomyopathy », Lebowitz et al provided an overview of the different treatments of hypertrophic obstructive cardiomyopathy (HOCM). This is a subject of particular interest in regard with the not negligeable incidence of HOCM in clinical practice, the impact of the disease on patients prognosis and the availability of different therapies. The paper is well-illustrated with some scheme and operative views.
My major concerns are as follow:
1. The authors should add a paragraph on medical treatment of HOCM. Only mavacamten is discussed in the current version of the paper. However, beta-blockers are use in clinical practice to reduce intraventricular gradient. Please precise the place of medical treatment in management of HOCM;
2. I think that the global plan of the review is quite difficult to follow for the readers. Please propose a more natural plan (for instance: medical treatment, validated interventional treatment, alternative interventional treatment, complications treatment, future directions);
3. The authors bring some interesting elements on ECMO in the field of HOCM. Please precise and discuss if it exists also data on other cardiocirculatory assistance;
4. The review contains a lot of abbreviations which make the reading difficult. For instance, I wonder whether it is necessary to use an abbreviation for “mitral valve” or “subvalvular apparatus”. Please amend.
I have several minor concerns:
· Abstract: please remove the term “compare” in the objectives of the review. This paper does not report the results of a comparative study, so the authors should use another term such as “discuss the place of the different therapeutic strategies” for example;
· Introduction part: please add some epidemiological data in regard with HOCM;
· Lines 527-529: Please remove the titles “3. Results” and “4. Discussion”.
Author Response
In this Review entitled « Review of Contemporary Treatment Approaches and Critical Aprisal of Guidelines on Hypertrophic Obstructive Cardiomyopathy », Lebowitz et al provided an overview of the different treatments of hypertrophic obstructive cardiomyopathy (HOCM). This is a subject of particular interest in regard with the not negligeable incidence of HOCM in clinical practice, the impact of the disease on patients prognosis and the availability of different therapies. The paper is well-illustrated with some scheme and operative views.
My major concerns are as follow:
Comment 1. The authors should add a paragraph on medical treatment of HOCM. Only mavacamten is discussed in the current version of the paper. However, beta-blockers are use in clinical practice to reduce intraventricular gradient. Please precise the place of medical treatment in management of HOCM;
Reply 1. Thank you for this comment. An according paragraph has now been rewritten to better reflect not only the emerging strategies for HOCM per sebut also symptoms treatment.
Changes 1. Please see paragraph (page 16; lines 517-37)
Comment 2. I think that the global plan of the review is quite difficult to follow for the readers. Please propose a more natural plan (for instance: medical treatment, validated interventional treatment, alternative interventional treatment, complications treatment, future directions);
Reply 2. We agree with the Reviewer and have now made an attempt to restructure the review accordingly by changing the headings in a more easy to follow way.
Changes 2. Throughout the MS
Comment 3. The authors bring some interesting elements on ECMO in the field of HOCM. Please precise and discuss if it exists also data on other cardiocirculatory assistance;
Reply 3. Thank you for this comment. The data on ECMO and other mechanical support is unfortunately lacking in the setting of HOCM. We have now added the information of single-centres experiences.
Changes 3. Please see paragraph (page 14; lines 444-460)
Comment 4. The review contains a lot of abbreviations which make the reading difficult. For instance, I wonder whether it is necessary to use an abbreviation for “mitral valve” or “subvalvular apparatus”. Please amend.
Reply 4. We are grateful for this remark and changed these accordingly
Changes 4 . Throughout the MS
I have several minor concerns:
Comment 5. Abstract: please remove the term “compare” in the objectives of the review. This paper does not report the results of a comparative study, so the authors should use another term such as “discuss the place of the different therapeutic strategies” for example;
Reply 5. We agree and rephrased accordingly
Changes Please see paragraph (page 1; lines 25)
Comment 6. Introduction part: please add some epidemiological data in regard with HOCM;
Reply 6. Thank you, epidemiological data is now added.
Changes 6. Please see paragraph (page 2; lines 45-47)
Comment 7. Lines 527-529: Please remove the titles “3. Results” and “4. Discussion”.
Reply 7. We are sorry for this imperfection, these lines are removed
Changes 7. Please see paragraph (page 17; lines 570)
Reviewer 2 Report
The authors present a very well-crafted review of HCOM. It is laid out well and progresses logically.
· Page 2, Lines 80-82. May want to comment on the fact that the sudden cardiac death risk does not have a correlation with gradient severity.
· Page 2, line 83, consider another term instead of safety backup such as prevention.
· Page 2, lines 84-85. You may want to include the anxiety and mental anguish, particularly over inappropriate shocks, as a complication of ICD.
· Page 3, lines 111-114. The wording is difficult to follow. It appears that > 30mmHg is in two risk categories. Review and consider revision.
· Page 3, line 119, and page 7, line 220. Consider an alternate word other than “vouched”
· Page 6 line 181 requires a reference.
· Page 7, line 218, define the massive septal thickness and youthful age. Consider expanding to provide some explanations as to why these factors resulted in a suboptimal response.
· Page 9, line 306-307. Consider adding a comparison to ASA for heart block to strengthen this section.
· Page 17, lines 527 and 529. Results and Discussion were left blank. Either add context or remove.
Author Response
Comment 1. The authors present a very well-crafted review of HCOM. It is laid out well and progresses logically.
Reply 1. Thank you for this comment, we appreciate your time revising our work
Changes 1. None
Comment 2. Page 2, Lines 80-82. May want to comment on the fact that the sudden cardiac death risk does not have a correlation with gradient severity.
Reply 2. We agree with the Reviewer and now insert this comment
Changes2 Please see paragraph (page 17; lines 570)
Comment 3. Page 2, line 83, consider another term instead of safety backup such as prevention.
Reply 3. We agree and deleted this phrase
Changes 3 Please see paragraph (page 2; lines 85
Comment 4. Page 2, lines 84-85. You may want to include the anxiety and mental anguish, particularly over inappropriate shocks, as a complication of ICD.
Reply 4. We agree and have now added these lines accordingly
Changes 4 Please see paragraph (page 2; lines 88)
Comment 5. Page 3, lines 111-114. The wording is difficult to follow. It appears that > 30mmHg is in two risk categories. Review and consider revision.
Reply 5. These are “provoked” and “resting” gradients categories; we have now rephrased for more clarity
Changes 5 Please see paragraph (page 3; lines 115-8)
Comment 6 Page 3, line 119, and page 7, line 220. Consider an alternate word other than “vouched”
Reply 6 We agree with the Reviewer and instated “demonstrated” instead
Changes 6 Please see paragraph (page 3; lines 122-3)
Comment 7 Page 6 line 181 requires a reference.
Reply 7 We are grateful for this remark, according figure is now referenced
Changes 7 Please see paragraph (page 7; lines 185)
Comment 8 Page 7, line 218, define the massive septal thickness and youthful age. Consider expanding to provide some explanations as to why these factors resulted in a suboptimal response.
Reply 8 We are grateful for this remark, these respective comments have now been added
Changes 8 Please see paragraph (page 7; lines 222-6)
Comment 9 Page 9, line 306-307. Consider adding a comparison to ASA for heart block to strengthen this section.
Reply 9; We are grateful for this comment; indeed the comparative studies are lacking, we have now added these respective lines.
Changes 9Please see paragraph (page 11; lines 324-5)
Comment 10 Page 17, lines 527 and 529. Results and Discussion were left blank. Either add context or remove.
Reply 10. We are sorry for this imperfection, these lines are removed
Changes 10. Please see paragraph (page 17; lines 570)